# REGRET MINIMIZATION FOR PARTIALLY OBSERVABLE DEEP REINFORCEMENT LEARNING

## ABSTRACT

Deep reinforcement learning algorithms that estimate state and state-action value functions have been shown to be effective in a variety of challenging domains, including learning control strategies from raw image pixels. However, algorithms that estimate state and state-action value functions typically assume a fully observed state and must compensate for partial or non-Markovian observations by using finite-length frame-history observations or recurrent networks. In this work, we propose a new deep reinforcement learning algorithm based on counterfactual regret minimization that iteratively updates an approximation to a cumulative clipped advantage function and is robust to partially observed state. We demonstrate that on several partially observed reinforcement learning tasks, this new class of algorithms can substantially outperform strong baseline methods: on Pong with single-frame observations, and on the challenging Doom (ViZDoom) and Minecraft (Malmö) first-person navigation benchmarks.

## 1 INTRODUCTION

Many reinforcement learning problems of practical interest have the property of partial observability, where observations of state are generally non-Markovian. Despite the importance of partial observation in the real world, value function-based methods such as Q-learning (Mnih et al., 2013; 2015) generally assume a Markovian observation space. On the other hand, Monte Carlo policy gradient methods do not assume Markovian observations, but many practical policy gradient methods such as A3C (Mnih et al., 2016) introduce the Markov assumption when using a critic or state-dependent baseline in order to improve sample efficiency.

Consider deep reinforcement learning methods that learn a state or state-action value function. One common workaround for the problem of partial observation is to learn value functions on the space of finite-length frame-history observations, under the assumption that frame-histories of sufficient length will give the environment the approximate appearance of full observability. When learning to play Atari 2600 games from images, deep Q-learning algorithms (Mnih et al., 2013; 2015) concatenate the last 4 observed frames of the video screen buffer as input to a state-action value convolutional network. Not all non-Markovian tasks are amenable to finite-length frame-histories; recurrent value functions can incorporate longer and potentially infinite histories (Hausknecht & Stone, 2017; Foerster et al., 2016), but at the cost of solving a harder optimization problem. Can we develop methods that learn a variant of the value function that is more robust to partial observability?

Our contribution is a new model-free deep reinforcement learning algorithm based on the principle of regret minimization which does not require access to a Markovian state. Our method learns a policy by estimating a *cumulative clipped advantage function*, which is an approximation to a type of regret that is central to two partial information game-solving algorithms from which we draw our primary inspiration: counterfactual regret minimization (CFR) (Zinkevich et al., 2007) and CFR+ (Tammelin, 2014). Hence we call our algorithm "advantage-based regret minimization" (ARM).

We evaluate our approach on three visual reinforcement learning domains: Pong with varying frame-history lengths (Bellemare et al., 2013), and the first-person games Doom (Kempka et al., 2016) and Minecraft (Johnson et al., 2016). Doom and Minecraft exhibit a first-person viewpoint in a 3-dimensional environment and should appear non-Markovian even with frame-history observations. We find that our method offers substantial improvement over prior methods in these partially observ-

able environments: on both Doom and Minecraft, our method can learn well-performing policies within about 1 million simulator steps using only visual input frame-history observations.

## 2 RELATED WORK

Deep reinforcement learning algorithms have been demonstrated to achieve excellent results on a range of complex tasks, including playing games (Mnih et al., 2015; Oh et al., 2016) and continuous control (Schulman et al., 2015; Lillicrap et al., 2016; Levine et al., 2016). Prior deep reinforcement learning algorithms either learn state or state-action value functions (Mnih et al., 2013), learn policies using policy gradients (Schulman et al., 2015), or perform a combination of the two using actor-critic architectures (Mnih et al., 2016). Policy gradient methods typically do not need to assume a Markovian state, but tend to suffer from poor sample complexity, due to their inability to use off-policy data. Methods based on learning Q-functions can use replay buffers to include off-policy data, accelerating learning (Lillicrap et al., 2016). However, learning Q-functions with Bellman error minimization typically requires a Markovian state space. When learning from observations such as images, the inputs might not be Markovian. Prior methods have proposed to mitigate this issue by using recurrent critics and Q-functions (Hausknecht & Stone, 2017; Oh et al., 2016; Mnih et al., 2016; Heess et al., 2015), and learning Q-functions that depend on entire histories of observations. Heuristics such as concatenation of short observation sequences have also been used (Mnih et al., 2015). However, all of these changes increase the size of the input space, increasing variance, and make the optimization problem more complex. Our method instead learns cumulative advantage functions that depend only on the current state, but can still handle non-Markovian problems.

The form of our advantage function update resembles positive temporal difference methods (Peng et al., 2016; van Hasselt & Wiering, 2007). Additionally, our update rule for a modified cumulative Q-function resembles the average Q-function (Anschel et al., 2017) used for variance reduction in Q-learning. In both cases, the theoretical foundations of our method are based on cumulative regret minimization, and the motivation is substantively different. Previous work by Ross et al. (2011); Ross & Bagnell (2014) has connected regret minimization to reinforcement learning, imitation learning, and structured prediction, although not with counterfactual regret minimization. Regression regret matching (Waugh et al., 2015) is based on a closely related idea, which is to directly approximate the regret with a linear regression model, however the use of a linear model is limited in representation compared to deep function approximation.

## 3 ADVANTAGE-BASED REGRET MINIMIZATION

In this section, we provide background on CFR and CFR+, describe ARM in detail, and give some intuition for why ARM works.

### 3.1 COUNTERFACTUAL REGRET MINIMIZATION (CFR)

In this section we review the algorithm of counterfactual regret minimization (Zinkevich et al., 2007). We closely follow the version of CFR as described in the Supplementary Material of Bowling et al. (2015), except that we try to use the notation of reinforcement learning where appropriate.

Consider the setting of an extensive game. There are $N$ players numbered $i = 1, \ldots, N$. An additional player may be considered a "chance" player to simulate random events. At each time step of the game, one player chooses an action $a \in \mathcal{A}_i$. Define the following concepts and notation:

- Sequences: A sequence specifically refers to a sequence of actions starting from an initial game state. (It is assumed that a sequence of actions, including actions of the "chance" player, is sufficient for defining state within the extensive game.) Let $\mathcal{H}$ be the space of all sequences, and let $\mathcal{Z}$ be the space of terminal sequences.

- Information sets: Let $\mathcal{I}$ be the space of information sets; that is, for each $I \in \mathcal{I}$, $I$ is a set of sequences $h \in I$ which are indistinguishable to the current player. Information sets are a represention of partial observability.

- Strategies: Let $\pi_i(a|I)$ be the strategy of the $i$-th player, where $\pi_i(a|I)$ is a probability distribution over action $a$ conditioned on information set $I$. Let $\pi = (\pi_1, \ldots, \pi_N)$ denote

the strategy profile for all players, and let $\pi_{-i} = (\pi_1, \ldots, \pi_{i-1}, \pi_{i+1}, \ldots, \pi_N)$ denote the strategy profile for all players except the $i$-th player.

- Sequence probabilities: Let $\rho^\pi(h)$ be the probability of reaching the sequence $h$ when all players follow $\pi$. Additionally, let $\rho^\pi(h, h')$ be the probability of reaching $h'$ conditioned on $h$ having already been reached. Similarly, define $\rho_i^\pi$ and $\rho_{-i}^\pi$ to contain the contributions of respectively only the $i$-th player or of all players except the $i$-th.

- Values: Let $u_i(z)$ be the value of a terminal sequence $z$ to the $i$-th player. Let the expected value of a strategy profile $\pi$ to the $i$-th player be $J_i(\pi) = \sum_{z \in Z} \rho^\pi(z) u_i(z)$.

Define the *counterfactual value* $Q_{\pi,i}^{\text{CF}}$ of all players following strategy $\pi$, except the $i$-th player plays to reach information set $I$ and to then take action $a$:

$$Q_{\pi,i}^{\text{CF}}(I, a) = \sum_{h \in I} \sum_{z \in \mathcal{Z}: h \sqsubset z} \rho_{-i}^\pi(z) \rho_i^{\pi|I \to a}(h, z) u_i(z). \tag{1}$$

The notation $h \sqsubset h'$ denotes that $h$ is a prefix of $h'$, while $\pi|I \to a$ denotes that action $a$ is to be performed when $I$ is observed. The counterfactual value $Q_{\pi,i}^{\text{CF}}(I, a)$ is a calculation that assumes the $i$-th player reaches any $h \in I$, and upon reaching any $h \in I$ it always chooses $a$.

Consider a learning scenario where at the $t$-th iteration the players follow a strategy profile $\pi^t$. The $i$-th player's regret after $T$ iterations is defined in terms of the $i$-th player's optimal strategy $\pi_i^*$:

$$R_i^T = \sum_{t=0}^{T-1} J_i((\pi_1^t, \ldots, \pi_{i-1}^t, \pi_i^*, \pi_{i+1}^t, \ldots, \pi_N^t)) - J_i(\pi^t). \tag{2}$$

The average regret is the average over learning iterations: $(1/T) R_i^T$. Now define the *counterfactual regret* of the $i$-th player for taking action $a$ at information set $I$:

$$(R_i^{\text{(CF)}})^T(I, a) = \sum_{t=0}^{T-1} \left( Q_{\pi^t, i}^{\text{CF}}(I, a) - \sum_{a' \in \mathcal{A}} \pi_i^t(a'|I) Q_{\pi^t, i}^{\text{CF}}(I, a') \right) \tag{3}$$

$$= (R_i^{\text{(CF)}})^{T-1}(I, a) + Q_{\pi^{T-1}, i}^{\text{CF}}(I, a) - \sum_{a' \in \mathcal{A}} \pi_i^{T-1}(a'|I) Q_{\pi^{T-1}, i}^{\text{CF}}(I, a'). \tag{4}$$

The counterfactual regret (Equation (3)) can be shown to majorize the regret (Equation (2)) (Theorem 3, Zinkevich et al. (2007)). CFR can then be described as a learning algorithm where the strategy is updated using regret matching (Hart & Mas-Colell, 2000) applied to the counterfactual regret calculated in the most recent iteration:

$$\pi_i^{T+1}(a|I) = \begin{cases} \frac{\max(0, (R_i^{\text{(CF)}})^{T+1}(I, a))}{\sum_{a' \in \mathcal{A}} \max(0, (R_i^{\text{(CF)}})^{T+1}(I, a'))} & \text{if } \sum_{a' \in \mathcal{A}} \max(0, (R_i^{\text{(CF)}})^{T+1}(I, a')) > 0 \\ \frac{1}{|A|} & \text{otherwise.} \end{cases} \tag{5}$$

If all players follow the CFR regret matching strategy (Equation (5)), then at the $T$-th iteration the players' average regrets are bounded by $O(T^{-1/2})$ (Theorem 4, Zinkevich et al. (2007)).

## 3.2 CFR+

CFR+ (Tammelin, 2014) consists of a modification to CFR, in which instead of calculating the full counterfactual regret as in (4), instead the counterfactual regret is recursively positively clipped to yield the *clipped* counterfactual regret:

$$(R_i^{\text{(CF+)}})^T(I, a) = \max(0, (R_i^{\text{(CF+)}})^{T-1}(I, a)) + Q_{\pi^{T-1}, i}^{\text{CF}}(I, a) - \sum_{a' \in \mathcal{A}} \pi_i^{T-1}(a'|I) Q_{\pi^{T-1}, i}^{\text{CF}}(I, a'). \tag{6}$$

Comparing Equation (4) with Equation (6), one can see that the only difference in CFR is that the previous iteration's counterfactual regret is positively clipped in the recursion. The one-line change of CFR+ turns out to yield a large practical improvement in the performance of the algorithm (Bowling et al., 2015), and there is also an associated regret bound for CFR+ that is as strong as the bound for CFR (Tammelin et al., 2015).

## 3.3 FROM CFR AND CFR+ TO ARM

CFR and CFR+ are formulated for imperfect information extensive-form games, so they are naturally generalized to partially observed stochastic games since a stochastic game can always be represented in extensive form. A 1-player partially observed stochastic game is simply a POMDP with observation space $\mathcal{O}$ (Littman, 1994). By mapping information sets $I \in \mathcal{I}$ to observations $o \in \mathcal{O}$, we may rewrite the counterfactual value as a kind of *stationary* observation-action value $Q^{\mathrm{CF}}_{\pi,i}(I, a) \equiv Q^{(\mathrm{stat})}_{\pi|o \mapsto a}(o, a)$ that assumes the agent follows the policy $\pi$ except on observing $o$, after which the action $a$ is always performed (Bellemare et al., 2016). We posit that the approximation $Q^{(\mathrm{stat})}_{\pi|o \mapsto a}(o, a) \approx Q_\pi(o, a)$, where $Q_\pi$ is the usual action value function, is valid when observations are rarely seen more than once in a trajectory. By approximating $Q^{(\mathrm{stat})}_{\pi|o \mapsto a}(o, a) \approx Q_\pi(o, a)$, we get a recurrence in terms of more familiar value functions (compare Equations (6) and (7)):

$$\bar{A}^+_t(o_k, a_k) = \max(0, \bar{A}^+_{t-1}(o_k, a_k)) + Q_{\pi_t}(o_k, a_k) - \sum_{a' \in \mathcal{A}} \pi_t(a'|o_k) Q_{\pi_t}(o_k, a') \tag{7}$$

$$= \max(0, \bar{A}^+_{t-1}(o_k, a_k)) + Q_{\pi_t}(o_k, a_k) - V_{\pi_t}(o_k) \tag{8}$$

$$= \max(0, \bar{A}^+_{t-1}(o_k, a_k)) + A_{\pi_t}(o_k, a_k) \tag{9}$$

where $\bar{A}^+_t(o, a)$ is the *cumulative clipped* advantage function, and $A_{\pi_t}(o, a)$ is the ordinary advantage function evaluated at policy $\pi_t$. Advantage-based regret minimization (ARM) is the resulting reinforcement learning algorithm that updates the policy to regret match on the cumulative clipped advantage function:

$$\pi_{t+1}(a_k|o_k) = \begin{cases} \frac{\max(0, \bar{A}^+_t(o_k, a_k))}{\sum_{a' \in \mathcal{A}} \max(0, \bar{A}^+_t(o_k, a'))} & \text{if } \sum_{a' \in \mathcal{A}} \max(0, \bar{A}^+_t(o_k, a')) > 0 \\ \frac{1}{|\mathcal{A}|} & \text{otherwise.} \end{cases} \tag{10}$$

Equations (9) and (10) suggest the outline of a batch-mode deep reinforcement learning algorithm. At the $t$-th sampling iteration, a batch of data is collected by sampling trajectories using the current policy $\pi_t$, followed by two processing steps: (a) fit $\bar{A}^+_t$ using Equation (9), then (b) set the next iteration's policy $\pi_{t+1}$ using Equation (10).

## 3.4 IMPLEMENTATION OF ARM

To implement Equation (9) with deep function approximation, we define two value function approximations, $V_{\pi_t}(o_k; \theta_t)$ and $\bar{Q}^+_t(o_k, a_k; \omega_t)$, as well as a target value function $V'(o_k; \varphi)$, where $\theta_t, \omega_t$, and $\varphi$ are the learnable parameters. The cumulative clipped advantage function is represented as $\bar{A}^+_t(o_k, a_k) = \bar{Q}^+_t(o_k, a_k; \omega_t) - V_{\pi_t}(o_k; \theta_t)$. Within each sampling iteration, the value functions are fitted using stochastic gradient descent by sampling minibatches and performing gradient steps. The state-value function $V_{\pi_t}(o_k; \theta_t)$ is fit to minimize an $n$-step temporal difference loss with a moving target $V'(o_{k+n}; \varphi)$, essentially using the estimator of the deep deterministic policy gradient (DDPG) (Lillicrap et al., 2016). In the same minibatch, $\bar{Q}^+_t(o_k, a_k; \theta_t)$ is fit to a similar loss, but with an additional target reward bonus that incorporates the previous iteration's cumulative clipped advantage, $\max(0, \bar{A}^+_{t-1}(o_k, a_k))$. The regression targets $v(o_k; \varphi)$ and $\bar{q}^+(o_k, a_k; \varphi)$ are defined in terms of the $n$-step returns $g^n_k = \sum_{k'=k}^{k+n-1} \gamma^{k'-k} r_{k'}$:

$$v(o_k; \varphi) \triangleq g^n_k + \gamma^n V'(o_{k+n}; \varphi) \tag{11}$$

$$q(o_k, a_k; \varphi) \triangleq r_k + \gamma g^{n-1}_{k+1} + \gamma^n V'(o_{k+n}; \varphi) \tag{12}$$

$$\bar{q}^+(o_k, a_k; \varphi) \triangleq \max(0, \bar{Q}^+_{t-1}(o_k, a_k; \omega_{t-1}) - V_{\pi_{t-1}}(o_k; \theta_{t-1})) + q(o_k, a_k; \varphi). \tag{13}$$

Altogether, each minibatch step of the optimization subproblem consists of the following three parameter updates in terms of the regression targets $v(o_k; \varphi)$ and $\bar{q}^+(o_k, a_k; \varphi)$:

$$\theta^{(\ell+1)}_t \leftarrow \theta^{(\ell)}_t - \frac{\alpha}{2} \nabla_{\theta^{(\ell)}_t} (V_{\pi_t}(o_k; \theta^{(\ell)}_t) - v(o_k; \varphi^{(\ell)}))^2 \tag{14}$$

$$\omega^{(\ell+1)}_t \leftarrow \omega^{(\ell)}_t - \frac{\alpha}{2} \nabla_{\omega^{(\ell)}_t} (\bar{Q}^+_t(o_k, a_k; \omega^{(\ell)}_t) - \bar{q}^+(o_k, a_k; \varphi^{(\ell)}))^2 \tag{15}$$

$$\varphi^{(\ell+1)} \leftarrow \varphi^{(\ell)} + \tau(\theta^{(\ell+1)}_t - \varphi^{(\ell)}). \tag{16}$$

---

**Algorithm 1** Advantage-based regret minimization (ARM).

---
initialize $\pi_0 \leftarrow$ uniform, $\theta_{-1}, \omega_{-1} \leftarrow$ arbitrary
**for** $t$ in $0, \ldots$ **do**
    collect batch of trajectory data $\mathcal{D}_t \sim \pi_t$
    initialize $\theta_t \leftarrow \theta_{t-1}, \omega_t \leftarrow \omega_{t-1}, \varphi \leftarrow \theta_{t-1}$
    **for** $\ell$ in $0, \ldots$ **do**
        sample transitions $(o_k, a_k, r_k, \ldots, o_{k+n-1}, a_{k+n-1}, r_{k+n-1}, o_{k+n}) \sim \mathcal{D}_t$
        calculate $n$-step returns $g_k^n = \sum_{k'=k}^{k+n-1} \gamma^{k'-k} r_{k'}$
        set $\delta_{k+n} \leftarrow \mathbb{I}[o_{k+n}$ is terminal$]$
        **if** $t = 0$ **then**
            set $\phi_k \leftarrow 0$
        **else**
            set $\phi_k \leftarrow \max(0, \bar{Q}_{t-1}^+(o_k, a_k; \omega_{t-1}) - V_{\pi_{t-1}}(o_k; \theta_{t-1}))$
        **end if**
        set $v(o_k) \leftarrow g_k^n + \gamma^n(1 - \delta_{k+n})V'(o_{k+n}; \varphi)$
        set $\bar{q}^+(o_k, a_k) \leftarrow \phi_k + r_k + \gamma g_{k+1}^{n-1} + \gamma^n(1 - \delta_{k+n})V'(o_{k+n}; \varphi)$
        update $\theta_t$ with step size $\alpha$ and targets $v(o_k)$ (Equation (14))
        update $\omega_t$ with step size $\alpha$ and targets $\bar{q}^+(o_k, a_k)$ (Equation (15))
        update $\varphi$ with moving average step size $\tau$ (Equation (16))
    **end for**
    set $\pi_{t+1}(a|o) \propto \max(0, \bar{Q}_t^+(o, a; \omega_t) - V_{\pi_t}(o; \theta_t))$
**end for**

---

The overall advantage-based regret minimization algorithm is summarized in Algorithm 1.

We note that the mechanics of the ARM updates are similar to on-policy value function estimation, but ARM learns a modified on-policy Q-function from transitions with the added reward bonus $\max(0, \bar{A}_{t-1}^+(o_k, a_k))$ (Equation (13)). This reward bonus can be thought of a kind of "optimism in the face of uncertainty."

## 3.5 ARM VS. EXISTING POLICY GRADIENT METHODS

In this section, we accentuate that ARM represents an inherently different update compared to existing policy gradient methods.

Recent work has shown that policy gradient methods and Q-learning methods are connected via entropy regularization (O'Donoghue et al., 2017; Haarnoja et al., 2017; Nachum et al., 2017; Schulman et al., 2017; Anonymous, 2018). One perspective is from the soft policy iteration framework for batch-mode reinforcement learning (Anonymous, 2018), where at each batch iteration the updated policy is obtained by minimizing the average KL-divergence between the policy class $\Pi$ and a target policy $f$. Below is the soft policy iteration update, where the subscript $t$ refers to the batch iteration:

$$\pi_{t+1} \leftarrow \arg\min_{\pi \in \Pi} \mathbb{E}_{o \sim \rho_t}[D_{\mathrm{KL}}(\pi \| f)] \tag{17}$$

$$= \arg\min_{\pi \in \Pi} \mathbb{E}_{o \sim \rho_t}[\mathbb{E}_{a \sim \pi(\cdot|o)}[\log(\pi(a|o)) - \log(f(a|o))]]. \tag{18}$$

Using the connection between policy gradient methods and Q-learning, we define the policy gradient target policy as the softmax distribution on the entropy regularized advantage function $A^{\beta\text{-soft}}$:

$$f^{\mathrm{PG}}(a|o) \triangleq \frac{\exp(\beta A_t^{\beta\text{-soft}}(o, a))}{\sum_{a' \in \mathcal{A}} \exp(\beta A_t^{\beta\text{-soft}}(o, a'))}. \tag{19}$$

We note that it is more conventional in the literature to use the soft Q-function $Q^{\beta\text{-soft}}(o, a)$ rather than the soft advantage function $A^{\beta\text{-soft}}(o, a)$, however since they differ only by a function of $o$ then they both induce the same target softmax policy. Now, parameterizing the policy $\pi$ in terms of an explicit parameter $\theta$, we obtain the expression for the existing policy gradient, where $b(o)$ is a baseline function:

$$\Delta\theta^{\mathrm{PG}} \propto \mathbb{E}_{o \sim \rho_t}[\mathbb{E}_{a \sim \pi(\cdot|o;\theta)}[\nabla_\theta \log(\pi(o|a;\theta))((1/\beta)\log(\pi(o|a;\theta)) - A_t^{\beta\text{-soft}}(o, a) + b(o))]]. \tag{20}$$

The classic policy gradient arises in the limit $\beta \to \infty$.

Note that an alternative choice of target policy $f$ will lead to a different kind of policy gradient update. A policy gradient algorithm based on ARM instead proposes the following target policy based on the regret-matching distribution:

$$f^{\text{ARM}}(a|o) \triangleq \frac{\max(0, \bar{A}_t^+(o, a))}{\sum_{a' \in \mathcal{A}} \max(0, \bar{A}_t^+(o, a'))}. \tag{21}$$

Similarly, we can express the ARM-like policy gradient, where again $b(o)$ is a baseline:

$$\Delta\theta^{\text{ARM}} = \mathbb{E}_{o \sim \rho_t}[\mathbb{E}_{a \sim \pi(\cdot|o;\theta)}[\nabla_\theta \log(\pi(o|a;\theta))(\log(\pi(o|a;\theta)) - \log(\max(0, \bar{A}_t^+(o, a))) + b(o))]]. \tag{22}$$

Comparing Equations (20) and (22), we see that the ARM-like policy gradient (Equation (22)) has a logarithmic dependence on the advantage-like function $\bar{A}^+$, whereas the existing policy gradient (Equation (20)) is only linearly dependent on the advantage function $A^{\beta\text{-soft}}$. This difference in logarithmic vs. linear dependence is responsible for a large part of the inherent distinction of ARM from existing policy gradient methods. One consequence of the difference in logarithmic vs. linear dependence is that the ARM-like update should be less sensitive to large positive advantages that may result from overestimation compared to existing policy gradient methods.

We also see that for the existing policy gradient (Equation (20)), the $(1/\beta) \log(\pi(a|o;\theta))$ term, which is derived from the policy entropy, is vanishing for large $\beta$ (e.g. $\beta = 100$ is a common choice in practice). On the other hand, for the ARM-like policy gradient (Equation (22)), there is no similar vanishing effect, suggesting that ARM may perform a kind of entropy regularization by default.

In practice we cannot implement an ARM-like policy gradient exactly as in Equation (22), as due to the positive clipping $\max(0, \bar{A}^+)$ there can appear $\log(0)$. However we believe this is not an intrinsic obstacle, leaving the issue of implementing an ARM-like policy gradient to future work.

### 3.6 WHY DOES ARM WORK BETTER IN PARTIALLY OBSERVABLE DOMAINS?

In the previous Section 3.5, we showed that ARM and existing policy gradient methods can be distinguished by their choices of target policy and the nature of their dependence on their respective advantage-like functions. In this section, we argue that the convergence results of CFR and CFR+ suggest that ARM, to the degree that it inherits the properties of CFR/CFR+, ought to benefit from greater partial observability compared to other methods.

We assume that regret bounds are a useful way to compare the convergence of different RL algorithms, due to the interpretation of regret as "area over the learning curve (and under the optimal expected value $J^*$)." Specifically, the regret bound of CFR and CFR+ is $O(|\mathcal{O}|\sqrt{T})$ where $|\mathcal{O}|$ is the size of the observation space (Zinkevich et al., 2007; Tammelin et al., 2015). The policy gradient method with a suitable baseline has a learning rate $\eta$-dependent regret bound derived from the stochastic gradient method; assuming parameter norm bound $B$ and gradient estimator second moments $G^2$, by setting the learning rate $\eta \propto T^{-1/2}$ policy gradient achieves a regret bound of $O(\sqrt{T})$ with no explicit dependence on the observation space size $|\mathcal{O}|$ (Dick, 2015).

We argue that possessing a regret bound proportional to the observation space size $|\mathcal{O}|$ is beneficial in highly partially observable domains. Let us fix an underlying state space $\mathcal{S}$. Compare two RL algorithms, where algorithm 1 (which is ARM-like) has a regret bound $c_1|\mathcal{O}|\sqrt{T}$, whereas algorithm 2 (which is policy gradient-like) has a regret bound $c_2\sqrt{T}$; here, $c_1$ and $c_2$ are constants. Note that if $c_1|\mathcal{O}| = c_2$ or equivalently $|\mathcal{O}| = c_2/c_1$, then the two RL algorithms possess the exact same regret bound. If on the other hand $|\mathcal{O}| < c_2/c_1$, then the regret bound of RL algorithm 1 is actually *lower* than that of RL algorithm 2. Applying this intuition to CFR and hence ARM suggests that ARM can benefit from greater partial observability if the degree of partial observability is above a threshold.

For Q-learning per se, we are not aware of any known regret bound. Szepesvári proved that the convergence rate of Q-learning in the $L^\infty$-norm, assuming a fixed exploration strategy, depends on a condition number $C$, which is the ratio of the minimum to maximum state-action occupation frequencies (Szepesvári, 1998), and which describes how "balanced" the exploration strategy is. If partial observability leads to imbalanced exploration due to confounding of states from perceptual aliasing (McCallum, 1997), then Q-learning should be negatively affected.

We note that there remains a gap between ARM as implemented and the theory of CFR: the use of (a) function approximation and sampling over tabular enumeration; (b) the "ordinary" Q-function instead of the "stationary" Q-function; and (c) $n$-step bootstrapped values instead of full returns for value function estimation. Waugh et al. (2015) address CFR with function approximation via a noisy version of a generalized Blackwell's condition (Cesa-Bianchi & Lugosi, 2003). Even the original implementation of CFR used sampling in place of enumeration (Zinkevich et al., 2007). We refer the reader to Bellemare et al. (2016) for a more in-depth discussion of the stationary Q-function. Although only the full returns are guaranteed to be unbiased in non-Markovian settings, it is quite common for practical RL algorithms to trade off strict unbiasedness in favor of lower variance by using $n$-step returns or variations thereof (Schulman et al., 2016; Gu et al., 2017).

## 4 EXPERIMENTS

Because we hypothesize that ARM should perform well in partially observable reinforcement learning environments, we conduct our experiments on visual domains that naturally provide partial observations of state. All of our evaluations use feedforward convnets with frame-history observations. We are interested in comparing ARM with methods that assume Markovian observations, namely double deep Q-learning (van Hasselt et al., 2016), as well as methods that can handle non-Markovian observations, primarily TRPO (Schulman et al., 2015; 2016), and to a lesser extent A3C (Mnih et al., 2016) whose critic assumes Markovian observations. We are also interested in controlling for the advantage structure of ARM by comparing with other advantage-structured methods, which include dueling networks (Wang et al., 2016), as well as policy gradient methods that estimate an empirical advantage using a baseline state-value function or critic (e.g. TRPO, A3C).

### 4.1 LEARNING TO PLAY PONG WITH A SINGLE FRAME

Atari games consist of a small set of moving sprites with fixed shapes and palettes, and the motion of sprites can be highly deterministic, so that with only 4 recently observed frames as input one can predict hundreds of frames into the future on some games using only a feedforward model (Oh et al., 2015). To increase the partial observability of Atari games, one may artificially limit the amount of frame history fed as input to the networks (Hausknecht & Stone, 2017). As a proof of concept of ARM, we trained agents to play Pong via the Arcade Learning Environment (Bellemare et al., 2013) when the frame-history length is varied between 4 (the default) and 1. We found that the performance of double deep Q-learning degraded noticeably when the frame-history length was reduced from 4 to 1, whereas performance of ARM was not affected nearly as much. Our results on Pong are summarized in Figure 1.

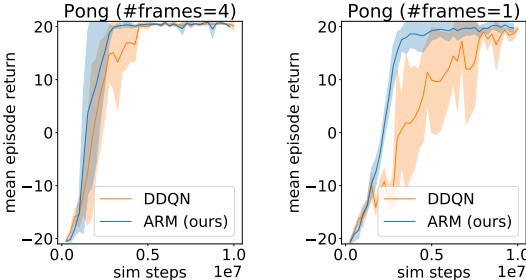

Figure 1: Comparing double deep Q-learning (orange) and ARM (blue) on Pong.

### 4.2 LEARNING TO NAVIGATE IN VIZDOOM

We evaluated ARM on the task of learning first-person navigation in the ViZDoom (Kempka et al., 2016) domain based on the game of Doom. Doom is a substantially more complex domain than Atari, featuring an egocentric viewpoint, 3D perspective, and complex visuals. We expect that Doom exhibits a substantial degree of partial observability and therefore serves as a more difficult evaluation of reinforcement learning algorithms' effectiveness on partially observable domains. We performed our evaluation on two standard ViZDoom navigation benchmarks, "HealthGathering"

and "MyWayHome." In "HealthGathering," the agent is placed in a toxic room and continually loses life points, but can navigate toward healthkit objects to prolong its life; the goal is to survive for as long as possible. In "MyWayHome," the agent is randomly placed in a small maze and must find a target object that has a fixed visual appearance and is in a fixed location in the maze; the goal is to reach the target object before time runs out. Figure 2 (top row) shows example observations from the two ViZDoom scenarios.

Unlike previous evaluations which augmented the raw pixel frames with extra information about the game state, e.g. elapsed time ticks or remaining health (Kempka et al., 2016; Dosovitskiy & Koltun, 2017), in our evaluation we forced all networks to learn using only visual input. Despite this restriction, ARM is still able to quickly learn policies with minimal tuning of hyperparameters and reach close to the maximum score in under 1 million steps. On "HealthGathering," we observed that ARM very quickly learns a policy that can achieve close to the maximum episode return of 2100. Double deep Q-learning learns a more consistent policy on "HealthGathering" compared to ARM and TRPO, but we believe this to be the result of evaluating double DQN's $\epsilon$-greedy policy with small $\epsilon$ compared to the truly stochastic policies learned by ARM and TRPO. On "MyWayHome," we observed that ARM generally learned a well-performing policy more quickly than other methods. Additionally, we found that ARM is able to take advantage of an off-policy replay memory when learning on ViZDoom by storing the trajectories of previous sampling batches and applying an importance sampling correction to the $n$-step returns; please see Section 6.2 in the Appendix for details. Our Doom results are in Figure 6.

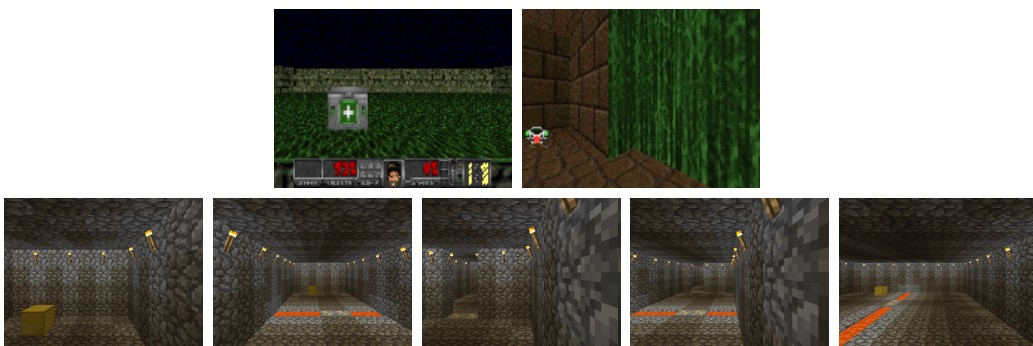

Figure 2: Top row: Doom screenshots from (left) "HealthGathering" and (right) "MyWayHome." Bottom row: Minecraft screenshots from (leftmost) "L1" through (rightmost) "L5"

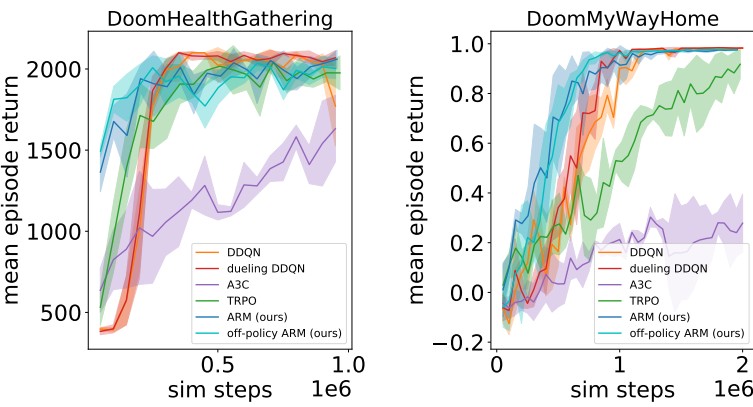

Figure 3: Evaluating double deep Q-learning (orange), dueling double DQN (red), A3C (purple), TRPO (green), ARM (blue), and ARM with off-policy data (cyan) on two ViZDoom scenarios.

### 4.3 LEARNING TO NAVIGATE IN MINECRAFT

We finally evaluated ARM on the task of learning first-person navigation in the Malmö domain based on the game of Minecraft (Johnson et al., 2016). Minecraft has similar visual complexity to Doom and should possess a comparable degree of partial observability, but Minecraft has the potential to be more difficult than Doom due to the diversity of possible Minecraft environments that can be generated. Our evaluation on Minecraft is adapted from the teacher-student curriculum learning protocol (Matiisen et al., 2017), which consists of 5 consecutive "levels" that successively increase the difficulty of completing the simple task of reaching a target block: the first level ("L1") consists of a single room; the intermediate levels ("L2"–"L4") consist of a corridor with lava-bridge and wall-gap obstacles; and the final level ("L5") consists of a $2 \times 2$ arrangement of rooms randomly separated by lava-bridge or wall-gap obstacles. Figure 2 (bottom row) shows example observations from the five Minecraft levels.

We performed our Minecraft experiments using fixed curriculum learning schedules to evaluate the sample efficiency of different algorithms: the agent is initially placed in the first level ("L1"), and the agent is advanced to the next level whenever a preselected number of simulator steps have elapsed, until the agent reaches the last level ("L5"). We found that ARM and dueling double DQN both were able to learn on an aggressive "fast" schedule of only 62500 simulator steps between levels. TRPO required a "slow" schedule of 93750 simulator steps between levels to reliably learn. ARM was able to consistently learn a well performing policy on all of the levels, whereas double DQN learned more slowly on some of the intermediate levels. ARM also more consistently reached a high score on the final, most difficult level ("L5"). Our Minecraft results are shown in Figure 4.

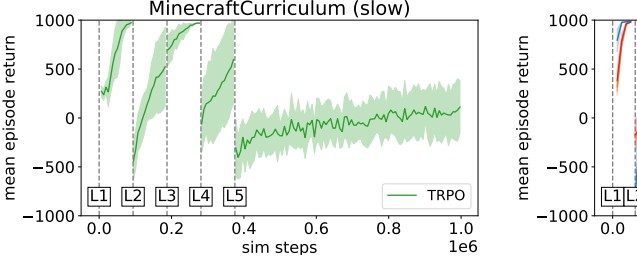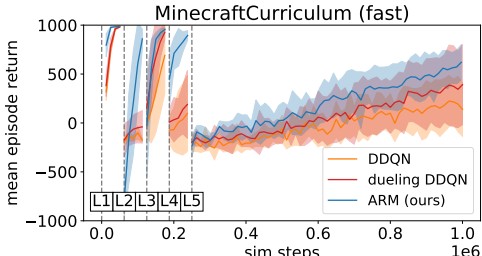

Figure 4: Evaluating double deep Q-learning (orange), dueling double DQN (red), TRPO (green), and ARM (blue) on a Minecraft curriculum learning protocol. The simulator step counts at which each level begins are labeled and demarcated with dashed vertical lines.

## 5 DISCUSSION

In this paper, we presented a novel deep reinforcement learning algorithm based on counterfactual regret minimization (CFR). We call our method advantage-based regret minimization (ARM). Similarly to prior methods that learn state or state-action value functions, our method learns a cumulative clipped advantage function of observation and action. However, in contrast to these prior methods, ARM is well suited to partially observed or non-Markovian environments, making it an appealing choice in a number of difficult domains. When compared to baseline methods, including deep Q-learning and TRPO, on non-Markovian tasks such as the challenging ViZDoom and Malmö first-person navigation benchmarks, ARM achieves substantially better results. This illustrates the value of ARM for partially observable problems. In future work, we plan to further explore applications of ARM to more complex tasks, including continuous action spaces.

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

## 6 Appendix

### 6.1 Experimental details

#### 6.1.1 Pong (Arcade Learning Environment)

We use the preprocessing and convolutional network model of (Mnih et al., 2013). Specifically, we view every 4th emulator frame, convert the raw frames to grayscale, and perform downsampling to generate a single observed frame. The input observation of the convnet is a concatenation of the most recent frames (either 4 frames or 1 frame). The convnet consists of an $8 \times 8$ convolution with stride 4 and 16 filters followed by ReLU, a $4 \times 4$ convolution with stride 2 and 32 filters followed by ReLU, a linear map with 256 filters followed by ReLU, and a linear map with $|\mathcal{A}|$ filters where $|\mathcal{A}|$ is the action space cardinality ($|\mathcal{A}| = 6$ for Pong).

We used Adam with a constant learning rate of $\alpha = 10^{-4}$, a minibatch size of 32, and the moment decay rates set to their defaults $\beta_1 = 0.9$ and $\beta_2 = 0.999$. Our results on each method are averaged across 3 random seeds.

We ran ARM with the hyperparameters: sampling batch size of 12500, 4000/3000 minibatches of Adam for the first/subsequent sampling iterations respectively, and target update step size $\tau = 0.01$. Double DQN uses the tuned hyperparameters (van Hasselt et al., 2016). Note that our choice of ARM hyperparameters yields an equivalent number of minibatch gradient updates per sample as used by DQN and double DQN, i.e. 1 minibatch gradient update per 4 simulator steps.

#### 6.1.2 Doom (ViZDoom)

We used a convolutional network architecture similar to those of (Kempka et al., 2016) and (Dosovitskiy & Koltun, 2017). The Doom screen was rendered at a resolution of $160 \times 120$ and downsized to $84 \times 84$. Only every 4th frame was rendered, and the input observation to the convnet is a concatenation of the last 4 rendered RGB frames for a total of 12 input channels. The convnet contains 3 convolutions with 32 filters each: the first is size $8 \times 8$ with stride 4, the second is size $4 \times 4$ with stride 2, and the third is size $3 \times 3$ with stride 1. The final convolution is followed by a linear map with 1024 filters. A second linear map yields the output. Hidden activations are gated by ReLUs.

For "HealthGathering" only, we scaled rewards by a factor of $0.01$. We did not scale rewards for "MyWayHome." We used Adam with a constant learning rate of $\alpha = 10^{-5}$ and a minibatch size of 32 to train all networks (except TRPO). For "HealthGathering" we set $\beta_1 = 0.95$, whereas for "MyWayHome" we set $\beta_1 = 0.9$. We set $\beta_2 = 0.999$ for both scenarios. Our results on each method are averaged across 3 random seeds.

Double DQN and dueling double DQN: $n = 5$ step returns; update interval 30000; 1 minibatch gradient update per 4 simulator steps; replay memory uniform initialization size 50000; replay memory maximum size 240000; exploration period 240000; with final exploration rate $\epsilon = 0.01$.

A3C: 16 workers; $n = 20$ steps for "HealthGathering" and $n = 40$ steps for "MyWayHome"; negentropy regularization $\beta = 0.01$; and gradient norm clip 5.

TRPO: sampling batch size 12500; KL-divergence step size $\delta = 0.01$; 10 conjugate gradient iterations; and Fisher information/Gauss-Newton damping coefficient $\lambda = 0.1$.

ARM: $n = 5$ step returns; sampling batch size 12500; 4000 Adam minibatches in the first sampling iteration, 3000 Adam minibatches in all subsequent sampling iterations; target update step size $\tau = 0.01$. Again, our choice of ARM hyperparameters yields an equivalent number of minibatch gradient updates per sample as used by DQN and double DQN. For "HealthGathering" only, because ARM converges so quickly we annealed the Adam learning rate to $\alpha = 2.5 \times 10^{-6}$ after 500000 elapsed simulator steps.

Off-policy ARM: $n = 5$ step returns; sampling batch size 1563, replay cache sample size 25000; 400 Adam minibatches per sampling iteration; target update step size $\tau = 0.01$; and importance sampling weight clip $c = 1$.

### 6.1.3 Minecraft (Malmö)

Our Minecraft tasks generally were the same as the ones used by Matiisen et al. (2017), with a few differences. Instead of using a continuous action space, we used a discrete action space with 4 move and turn actions. To aid learning on the last level ("L5"), we removed the reward penalty upon episode timeout and we increased the timeout on "L5" from 45 seconds to 75 seconds due to the larger size of the environment. We scaled rewards for all levels by $0.001$.

We use the same convolutional network architecture for Minecraft as we used for ViZDoom in Section 4.2. The Minecraft screen was rendered at a resolution of $320 \times 240$ and downsized to $84 \times 84$. Only every 5th frame was rendered, and the input observation of the convnet is a concatenation of the last 4 rendered RGB frames for a total of 12 input channels. We used Adam with constant learning rate $\alpha = 10^{-5}$, moment decay rates $\beta_1 = 0.9$ and $\beta_2 = 0.999$, and minibatch size 32 to train all networks (except TRPO). Our results on each method are averaged across 5 random seeds.

Double DQN and dueling double DQN: $n = 5$ step returns; update interval 12500; 1 minibatch gradient update per 4 simulator steps; replay memory uniform initialization size 12500; replay memory maximum size 62500; exploration period 62500; with final exploration rate $\epsilon = 0.01$.

TRPO: sampling batch size 6250; KL-divergence step size $\delta = 0.01$; 10 conjugate gradient iterations; and Fisher information/Gauss-Newton damping coefficient $\lambda = 0.1$.

ARM: $n = 5$ step returns; sampling batch size 12500; 4000 Adam minibatches in the first sampling iteration, 3000 Adam minibatches in all subsequent sampling iterations; target update step size $\tau = 0.01$.

### 6.2 Off-policy ARM via importance sampling

Our current approach to running ARM with off-policy data consists of applying an importance sampling correction directly to the $n$-step returns. Given the behavior policy $\mu$ under which the data was sampled, the current policy $\pi_t$ under which we want to perform estimation, and an importance sampling weight clip $c$ for variance reduction, the corrected $n$-step return we use is:

$$g_k^n(\mu\|\pi_t) = \sum_{k'=k}^{k+n-1} \gamma^{k'-k} \left( \prod_{\ell=k}^{k'} w_{\mu\|\pi_t}(a_\ell|o_\ell) \right) r_{k'} \tag{23}$$

where the truncated importance weight $w_{\mu\|\pi_t}(a|o)$ is defined (Ionides, 2008):

$$w_{\mu\|\pi_t}(a|o) = \min\left( c, \frac{\pi_t(a|o)}{\mu(a|o)} \right). \tag{24}$$

Our choice of $c = 1$ in our experiments was inspired by Wang et al. (2017). We found that $c = 1$ worked well but note other choices for $c$ may also be reasonable.

When applying our importance sampling correction, we preserve all details of the ARM algorithm except for two aspects: the transition sampling strategy (a finite memory of previous batches are cached and uniformly sampled) and the regression targets for learning the value functions. Specifically, the regression targets $v(o_k; \varphi)$, $q(o_k, a_k; \varphi)$, and $\bar{q}^+(o_k, a_k; \varphi)$ (Equations (11)–(13)) are modified to the following:

$$v_{\mu\|\pi_t}(o_k; \varphi) = g_k^n(\mu\|\pi_t) + \gamma^n V'(o_{k+n}; \varphi) \tag{25}$$

$$q_{\mu\|\pi_t}(o_k, a_k; \varphi) = r_k + \gamma w_{\mu\|\pi_t}(a_k|o_k) g_{k+1}^{n-1}(\mu\|\pi_t) + \gamma^n V'(o_{k+n}; \varphi) \tag{26}$$

$$\bar{q}_{\mu\|\pi_t}^+(o_k, a_k; \varphi) = \max(0, \bar{Q}_{t-1}^+(o_k, a_k; \omega_{t-1}) - V_{\pi_{t-1}}(o_k; \theta_{t-1})) + q_{\mu\|\pi_t}(o_k, a_k; \varphi). \tag{27}$$

Note that the target value function $V'(o_{k+n}; \varphi)$ does not require an importance sampling correction because $V'$ already approximates the on-policy value function $V_{\pi_t}(o_{k+n}; \theta_t)$.

### 6.3 Additional experiments

### 6.3.1 Atari 2600 games

Although our primary interest is in partially observable reinforcement learning domains, we also want to check that ARM works in nearly fully observable and Markovian environments, such as

Atari 2600 games. We consider two baselines: double deep Q-learning, and double deep fitted Q-iteration which is a batch counterpart to double DQN. We find that double deep Q-learning is a strong baseline for learning to play Atari games, although ARM still successfully learns interesting policies. One major benefit of Q-learning-based methods is the ability to utilize a large off-policy replay memory. Our results on a suite of Atari games are in Figure 5.

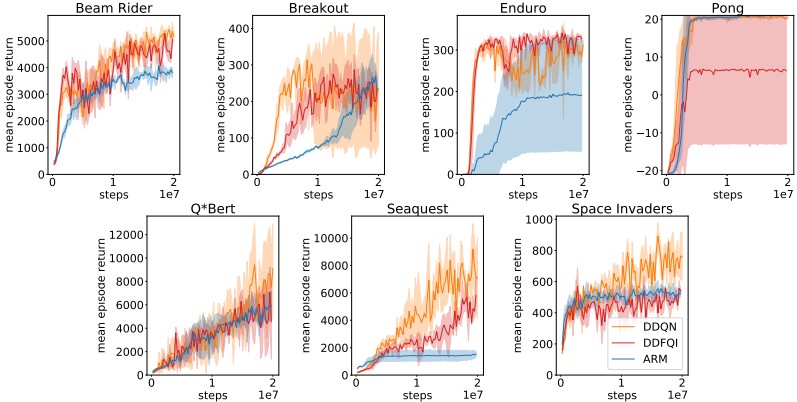

Figure 5: Comparing double deep Q-learning (orange), double deep fitted Q-iteration (red), and ARM (blue) on a suite of seven Atari games from the Arcade Learning Environment. For each method, we plot the mean across 3 trials along with standard error bars.

### 6.3.2 RECURRENCE IN DOOM MYWAYHOME

We evaluated the effect of recurrent policy and value function estimation in the maze-like MyWay-Home scenario of ViZDoom. We found that recurrence has a small positive effect on the convergence of A2C (Mnih et al., 2016), but was much less significant than the choice of algorithm. Our hyper-parameters were similar to those described for A3C in Section 6.1.2, except we used a learning rate $10^{-4}$ and gradient norm clip $0.5$. For the recurrent policy and value function, we replaced the first fully connected operation with an LSTM featuring an equivalent number of hidden units (1024).

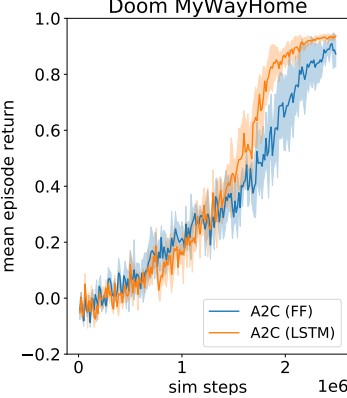

Figure 6: Comparing A2C with a feedforward convolutional network (blue) and a recurrent convolutional-LSTM network (orange) on the ViZDoom scenario MyWayHome.

