# OpenReview forum: "Regret Minimization for Partially Observable Deep Reinforcement Learning"
_ICLR.cc/2018/Conference — Invite to Workshop Track_

### Official Review · AnonReviewer1 · 2017-11-27
**A regret minimization approach to policy gradient algorithms.**

**Rating:** 4
**Confidence:** 4

**Review:**

This paper presents Advantage-based Regret Minimization, somewhat similar to advantage actor-critic with REINFORCE.
The main focus of the paper seems to be the motivation/justification of this algorithm with connection to the regret minimization literature (and without Markov assumptions).
The claim that ARM is more robust to partially observable domains is supported by experiments where it outperforms DQN.

There are several things to like about this paper:
- The authors do a good job of reviewing/referencing several papers in the field of "regret minimization" that would probably be of interest to the ICLR community + provide non-obvious connections / summaries of these perspectives.
- The issue of partial observability is good to bring up, rather than simply relying on the MDP framework that is often taken as a given in "deep reinforcement learning".
- The experimental results show that ARM outperforms DQN on a suite of deep RL tasks.

However, there are also some negatives:
- Reviewing so much of the CFR-literature in a short paper means that it ends up feeling a little rushed and confused.
- The ultimate algorithm *seems* like it is really quite similar to other policy gradient methods such as A3C, TRPO etc. At a high enough level, these algorithms can be written the same way... there are undoubtedly some key differences in how they behave, but it's not spelled out to the reader and I think the connections can be missed.
- The experiment/motivation I found most compelling was 4.1 (since it clearly matches the issue of partial observability) but we only see results compared to DQN... it feels like you don't put a compelling case for the non-Markovian benefits of ARM vs other policy gradient methods. Yes A3C and TRPO seem like they perform very poorly compared to ARM... but I'm left wondering how/why?

I feel like this paper is in a difficult position of trying to cover a lot of material/experiments in too short a paper.
A lot of the cited literature was also new to me, so it could be that I'm missing something about why this is so interesting.
However, I came away from this paper quite uncertain about the real benefits/differences of ARM versus other similar policy gradient methods... I also didn't feel the experimental evaluations drove a clear message except "ARM did better than all other methods on these experiments"... I'd want to understand how/why and whether we should expect this universally.
The focus on "regret minimization perspectives" didn't really get me too excited...

Overall I would vote against acceptance for this version.

---

> ### Author Response · Authors · 2018-01-05
> **Thanks for your comments**
>
> Thanks for your comments! We have addressed the clarity issues raised by the reviewers and added additional experiments to address reviewer concerns, which we detailed below.
>
> - Reviewing so much of the CFR-literature in a short paper means that it ends up feeling a little rushed and confused.
>
> We admit Section 3 can be cleaned up -- in the final we will address this.
>
> - The ultimate algorithm *seems* like it is really quite similar to other policy gradient methods such as A3C, TRPO etc.
>
> In our modified section 3.5 we address this point in more detail.
>
> One way to think about existing policy gradient methods is that they approximately minimize the KL-divergence between the policy and a target Boltzmann distribution induced either by the Q-function or by the advantage function. (For simplicity we ignore considerations of entropy regularization.) As a result of the KL-divergence/Boltzmann interpretation, the policy gradient update is linearly proportional to the advantage.
>
> By analogy, if one were to minimize the KL-divergence between the policy and a different kind of target distribution, the resulting gradient updates would also be different. In particular, ARM proposes a distribution based on regret-matching which is proportional to the positively clipped part of the cumulative clipped advantage (i.e. the "A+" function). If we were to implement a policy gradient-like version of ARM, then the resulting gradient update would be proportional to the _logarithm_ of the cumulative clipped advantage, and is inherently different than the existing policy gradient update. One consequence is that logarithmic dependence on the advantage means that ARM may be less sensitive to value function overestimation that may result in large positive advantages.
>
> - it feels like you don't put a compelling case for the non-Markovian benefits of ARM vs other policy gradient methods ... I'd want to understand how/why and whether we should expect this universally
>
> We added section 3.6 to address this point in more detail.
>
> In Section 3.6 we make an informal argument based on the regret bounds of CFR/CFR+ vs the convergence rates of other methods. For CFR/CFR+, the regret bound is proportional to the size of the observation space. On the other hand, for the policy gradient method, the regret bound has no direct dependence on the size of the observation space. This suggests that there could be a "threshold" level of the observation space size, such that a smaller observation space size (i.e. more partially observable) leads to better relative performance of CFR/CFR+ (and hence ARM), whereas a larger observation space size (i.e. more fully observable) leads to worse relative performance of CFR/CFR+ (and hence ARM).
>
> This argument suggests that ARM could outperform other methods when there is a high degree of partial observability in a domain (e.g. Minecraft). Conversely, when a domain is nearly fully observable (e.g. Atari) it is possible for ARM to converge slower than other methods -- these match our empirical results on Atari in the Appendix, section 6.3.1.
>
> Finally, there is also a lot of work showing that Q-learning based methods (i.e. DQN) can be much more sample efficient than policy gradient methods (e.g. A3C and TRPO). Estimating something that looks like a Q-function often leads to faster convergence. A deeper reason is that policy gradient methods are better at avoiding Markov assumptions by reducing dependence on value function estimation, but at the cost of sample efficiency.

---

### Official Review · AnonReviewer3 · 2017-12-01
**Interesting results on applying counterfactual regret minimization results to Deep RL.**

**Rating:** 7
**Confidence:** 4

**Review:**

This paper introduces the concepts of counterfactual regret minimization in the field of Deep RL. Specifically, the authors introduce an algorithm called ARM which can deal with partial observability better. The results is interesting and novel. This paper should be accepted.

The presentation of the paper can be improved a bit. Much of the notation introduced in section 3.1 is not used later on. There seems to be a bit of a disconnect before and after section 3.3. The algorithm in deep RL could be explained a bit better.

There are some papers that could be connected. Notably the distributional RL work that was recently published could be very interesting to compare against in partially observed environments.

It could also be interesting if the authors were to run the proposed algorithm on environments where long-term memory is required to achieve the goals.

The argument the authors made against recurrent value functions is that recurrent value could be hard to train. An experiment illustrating this effect could be illuminating.

Can the proposed approach help when we have recurrent value functions? Since recurrence does not guarantee that all information needed is captured.


Finally some miscellaneous points:

One interesting reference: Memory-based control with recurrent neural
networks by Heess et al.

Potential typos: in the 4th bullet point in section 3.1, should it be \rho^{\pi}(h, s')?

---

> ### Author Response · Authors · 2018-01-05
> **Thanks for your comments**
>
> Thanks for your comments! We have addressed the clarity issues raised by the reviewers and added additional experiments to address reviewer concerns, which we detailed below.
>
> - The presentation of the paper can be improved a bit
>
> We admit Section 3 can be cleaned up -- in the final we will address this.
>
> - There are some papers that could be connected. Notably the distributional RL work that was recently published could be very interesting to compare against in partially observed environments. ... One interesting reference: Memory-based control with recurrent neural networks by Heess et al.
>
> Thanks! In the final we will elaborate on additional connections with the literature.
>
> - The argument the authors made against recurrent value functions is that recurrent value could be hard to train. An experiment illustrating this effect could be illuminating.
>
> We added Section 6.3.2 with an experiment to illustrate this effect.
>
> We compared feedforward and recurrent convolutional policies and value functions learned using A2C on the maze-like ViZDoom MyWayHome scenario. Adding recurrence does seem to have a small positive effect, but it is less than the effect due to e.g. choice of algorithm.
>
> - Can the proposed approach help when we have recurrent value functions? Since recurrence does not guarantee that all information needed is captured.
>
> As ARM only involves different value function estimators it should be able to handle recurrent value functions. In practice we currently run ARM in batch mode only. An online version of ARM would handle recurrence much more naturally; this work is in progress.
>
> - Potential typos: in the 4th bullet point in section 3.1, should it be \rho^{\pi}(h, s')?
>
> Thanks, this was a typo, it should be: s' -> h'

---

### Official Review · AnonReviewer2 · 2017-12-01
**The paper provides a game-theoretic inspired variant of policy-gradient algorithm based on the idea of counter-factual regret minimization. The paper claims that the approach can deal with the partial observable domain better than the standard methods. However the results only show that the algorithm converges, in some cases, faster than the previous work.**

**Rating:** 5
**Confidence:** 5

**Review:**

Quality and clarity:

The paper provides a game-theoretic inspired variant of policy-gradient algorithm based on the idea of counter-factual regret minimization. The paper claims that the approach can deal with the partial observable domain better than the standard methods. However the results only show that the algorithm converges, in some cases, faster than the previous work  reaching asymptotically to a same or worse performance. Whereas one would expect that the algorithm achieve a better asymptotic performance in compare to methods which are designed for fully observable domains and thus performs sub-optimally in the POMDPs.

The paper dives into the literature of counter-factual regret minimization without providing much intuition on why this type of ideas should provide improvement in the case of partial observable domain. To me it is not clear at all why this idea should help in the partial observable domains beside the argument that this method is designed in the game-theoretic settings   which makes no Markov assumption . The way that I interpret this algorithm is that by adding A+ to the return the algorithm  introduces some bias for actions which are likely to be optimal so it is in some sense implements the optimism in the face of uncertainty principle. This may explains why this algorithm converges faster than the baseline as it produces better exploration strategy. To me it is not clear that the boost comes from the fact that the algorithm deals with partial observability more efficiently.


Originality and Significance:

The proposed algorithm seems original. However,  as it is acknowledged by the authors this type of optimistic policy gradient algorithms have been previously used in RL (though maybe not with the game theoretic justification). I believe the algorithm introduced  in this paper, if it is presented well, can be  an interesting addition to the literature of Deep RL, e.g.,  in terms of improving the rate of convergence. However, the current version of paper  does not provide conclusive evidence for that as in most of the domains the algorithm only converge marginally faster than the standard ones. Given the fact that algorithms like dueling DQN and DDPG are   for the best asymptotic results and not  for the best convergence rate, this improvement  can be due to the choice of hyper parameter such as step size or epsilon decay scheduling. More experiments over a range of hyper parameter is needed before one can conclude that this algorithm improves the rate of convergence.

---

> ### Author Response · Authors · 2018-01-05
> **Thanks for your comments**
>
> Thanks for your comments! We have addressed the clarity issues raised by the reviewers and added additional experiments to address reviewer concerns, which we detailed below.
>
> - The paper dives into the literature of counter-factual regret minimization without providing much intuition on why this type of ideas should provide improvement in the case of partial observable domain. To me it is not clear at all why this idea should help in the partial observable domains beside the argument that this method is designed in the game-theoretic settings  which makes no Markov assumption .
>
> We have added Section 3.6 with this information.
>
> In Section 3.6 we make an informal argument based on the regret bounds of CFR/CFR+ vs the convergence rates of other methods. For CFR/CFR+, the regret bound is proportional to the size of the observation space. On the other hand, for the policy gradient method, the regret bound has no direct dependence on the size of the observation space. This suggests that there could be a "threshold" level of the observation space size, such that a smaller observation space size (i.e. more partially observable) leads to better relative performance of CFR/CFR+ (and hence ARM), whereas a larger observation space size (i.e. more fully observable) leads to worse relative performance of CFR/CFR+ (and hence ARM).
>
> This argument suggests that ARM could outperform other methods when there is a high degree of partial observability in a domain (e.g. Minecraft). Conversely, when a domain is nearly fully observable (e.g. Atari) it is possible for ARM to converge slower than other methods -- these match our empirical results on Atari in the Appendix, section 6.3.1.
>
> - adding A+ to the return the algorithm  introduces some bias for actions which are likely to be optimal so it is in some sense implements the optimism in the face of uncertainty principle.
>
> Thanks! This is a good point which we added to our modified Section 3.4.
>
> - this type of optimistic policy gradient algorithms have been previously used in RL (though maybe not with the game theoretic justification)
>
> In our modified section 3.5 we address this point in more detail.
>
> One way to think about existing policy gradient methods is that they approximately minimize the KL-divergence between the policy and a target Boltzmann distribution induced either by the Q-function or by the advantage function. (For simplicity we ignore considerations of entropy regularization.) As a result of the KL-divergence/Boltzmann interpretation, the policy gradient update is linearly proportional to the advantage.
>
> By analogy, if one were to minimize the KL-divergence between the policy and a different kind of target distribution, the resulting gradient updates would also be different. In particular, ARM proposes a distribution based on regret-matching which is proportional to the positively clipped part of the cumulative clipped advantage (i.e. the "A+" function). If we were to implement a policy gradient-like version of ARM, then the resulting gradient update would be proportional to the _logarithm_ of the cumulative clipped advantage, and is inherently different than the existing policy gradient update. One consequence is that logarithmic dependence on the advantage means that ARM may be less sensitive to value function overestimation that may result in large positive advantages.
>
> - algorithms like dueling DQN and DDPG are for the best asymptotic results and not for the best convergence rate
>
> Regret in the context of RL can be thought of as "area over the learning curve (and under the optimal expected return)". In other words, regret is a measure of sample efficiency. Faster minimization of regret leads to faster convergence rate but generally does not change the overall asymptotic performance that can be attained. Empirically we observe that ARM does achieve higher performance within a finite number of steps on some tasks compared to others (the Minecraft one in particular).
>
> We are happy to compare with any other methods, but to our knowledge dueling double DQN is a strong baseline for the empirical convergence rate in addition to the asymptotic performance (we didn't compare with DDPG because we only evaluated discrete action space domains).
>
> - More experiments over a range of hyper parameter is needed before one can conclude that this algorithm improves the rate of convergence
>
> Some general comments about our hyperparameter tuning:
> (a) by fixing the network architecture, we found that the learning rate and other optimizer hyperparams were mostly architecture-dependent (except for TRPO);
> (b) the number of steps to use for n-step returns made the biggest difference and was independent of algorithm.

---

### Public Comment · (anonymous) · 2018-01-04
**Regret of POMDPs**

Hi authors,

The title of your paper reminds me a theory paper on regret bound of POMDPs.
"Reinforcement learning of POMDPs using spectral methods" which does regret minimization of POMDPS.
I skimmed your paper a bit, and I think it would good to discuss this paper.

---

> ### Author Response · Authors · 2018-01-05
> **Spectral methods and model-based RL**
>
> Thanks for your comment. In our work we only considered model-free deep RL methods, so I'm not super familiar with the SM-UCRL work by Azizzadenesheli et al which involves POMDP model estimation. My initial impression though is that estimation of the POMDP model parameters using spectral methods is an interesting idea for model-based RL in general.

---

### Decision · Program_Chairs · 2018-01-29
**ICLR 2018 Conference Acceptance Decision**

**Decision:**

Invite to Workshop Track

**Comment:**

The reviewers agree this is a really interesting paper, with an interesting idea (in particular
the use of regret clipping might provide a benefit over typical policy gradient methods). However,
there are two major concerns: 1) clarity / exposition and more importantly 2) lack of a strong
empirical motivation for the new approach (why do standard methods work just as well on these
partially observable domains?).